# Research Update of Emergent Sulfur Quantum Dots in Synthesis and Sensing/Bioimaging Applications

**DOI:** 10.3390/molecules27092822

**Published:** 2022-04-28

**Authors:** Keke Ning, Yujie Sun, Jiaxin Liu, Yao Fu, Kang Ye, Jiangong Liang, Yuan Wu

**Affiliations:** College of Science, Huazhong Agricultural University, Wuhan 430070, China; queenkerrning@outlook.com (K.N.); syj01234560123@163.com (Y.S.); ljx1280270429@163.com (J.L.); fy1875241435@163.com (Y.F.); yekang0322@outlook.com (K.Y.)

**Keywords:** sulfur quantum dots, synthesis, sensing applications, challenges

## Abstract

Due to their unique optical property, low toxicity, high hydrophilicity, and low cost, sulfur quantum dots (SQDs), an emerging luminescent nanomaterial, have shown great potential in various application fields, such as sensing, bioimaging, light emitting diode, catalysis, and anti-bacteria. This minireview updates the synthetic methods and sensing/bioimaging applications of SQDs in the last few years, followed by discussion of the potential challenges and prospects in their synthesis and sensing/bioimaging applications, with the purpose to provide some useful information for researchers in this field.

## 1. Introduction

In the past few years, semiconductor quantum dots (QDs), fluorescent nanocrystals with a size smaller than the exciton Bohr radius [1], have attracted wide interest from researchers of different fields, due to their excellent physical and chemical properties, such as high quantum yields (QY), high photostability, ultra-bright photoluminescence (PL), and tunability of emission wavelengths, making them particularly suitable for biological applications and imaging [2]. However, traditional QDs are usually composed of toxic heavy metal elements, such as cadmium (Cd) [3], lead (Pb) [4], and mercury (Hg) [5], which are shown to be seriously harmful to the environment and biological systems even at a low concentration [6]. Therefore, researchers have made persistent efforts in the past decade to explore heavy metal-free QDs with satisfactory biocompatibility, low or non-toxicity, and chemical inertness, such as carbon quantum dots (CDs) [7], graphene quantum dots (GQDs) [8], silicon quantum dots (SiQDs) [9], silver quantum dots (AgQDs) [10], and phosphorene quantum dots (PQDs) [11].

Elemental sulfur, one of the most abundant and extensively used substances on earth [12], used to be mainly extracted from deposits in volcanic rocks by chemical processes, such as smelting and refining a century ago [13]. In modern times, the global sulfur is mainly produced from crude oil refining process. The huge demand for sulfur in various fields, such as sulfuric acid production, medicine, rubber production, lithium–sulfur batteries, and agriculture [14,15,16,17]. Apart from meeting the huge demand for sulfur in various industries, a large amount of sulfur has not been fully utilized, resulting in a huge waste of sulfur resources. Considering the flammable and explosive characteristics of sulfur, it is urgent to increase the exploitation/development and utilization of residual elemental sulfur.

With the rapid development of modern nanotechnology, various types of sulfur nanomaterials have been developed, including sulfur nanoparticles (SNPs) [18], sulfur quantum dots (SQDs) [19], and other core/shell, porous, hybrid, and assembly nanostructures [20,21,22,23,24]. Since their first report in 2014 [19], SQDs have attracted much interest as emerging and active metal-free elemental QDs, due to their special properties different from bulk sulfur, such as facile synthesis, low toxicity, and good optical properties. Despite their research in the initial stage, SQDs have been applied in various fields, such as sensing [25], imaging [26], photocatalysis [24], fabricating light emitting device [27], anti-bacteria [28], and so on [29].

In this minireview, we summarize the synthetic methods of SQDs from the perspective of precursors, solvent, ligand, temperature, reaction time, QY, emission wavelength, and reaction yield (Table 1) as well as their applications in sensing and imaging, whereas sensing including fluorescence sensing, colorimetric sensing, ratiometric sensing, electrochemical sensing, and electrochemiluminescence (ECL) sensing (Figure 1). Additionally, we discuss the challenges and prospects of SQDs in synthesis and sensing/bioimaging applications.

## 2. Synthetic Methods of SQDs

Currently, the synthetic methods of SQDs mainly include two strategies: (i) acid etching oxidation of metal sulfide QDs, such as CdS and ZnS QDs, and (ii) the top-down method using elemental sulfur to synthesize SQDs, mainly including the assembly-fission method, surface-etching method, ultrasonication and microwave method, oxygen-accelerated method, one-step hydrothermal method, and so on.

### 2.1. Acid Etching Oxidation Method

In 2014, Li et al. first reported the acid etching oxidation method of using CdS and ZnS QDs as precursors to prepare luminescent SQDs with an average size of 1.6 nm as well as 428 nm of emission under 352 nm excitation light [19]. As shown in Figure 1, this method mainly included the steps of physical contact, oil–water phase interfacial reaction, in situ precipitation and dissolution, especially the addition of HNO_3_, enabling the S^2−^ present in CdS QDs (2.9 nm of size) to be slowly oxidized to element sulfur, etching and dissolving Cd^2+^ in the solution to obtain SQDs. The obtained SQDs possess the advantages of high hydrophilicity, abundant surface functional groups, excitation-dependent photoluminescence, high photostability, and low toxicity. However, they also suffer from the drawbacks of low QY (0.549% relative to quinine sulfate), long reaction time (36 h from synthesis of CdS QDs to obtain SQDs), complex operation, harsh reaction conditions, and high cost; thus seriously limiting the large-scale preparation of SQDs and their wide applications. These drawbacks motivated researchers to further develop facile and low-cost methods for large-scale synthesis of SQDs.

### 2.2. Assembly-Fission Method

In 2018, four years after the first report of SQDs, Shen’s group reported the synthesis of luminescent SQDs with the assembly-fission method, using sublimated sulfur powders as precursors and polyethylene glycol-400 (PEG-400) as the stabilizer under an alkaline condition. The whole process was divided into the three steps of dissolution, assembly, and fission (Figure 2A) [30]. As shown in Figure 2B, the sulfur powders are firstly dissolved in alkaline solution (NaOH) by inches to generate sodium sulfide (Na_2_S), followed by the reaction between Na_2_S and sulfur powders to produce sodium polysulfide. During the first 30 h of reaction, bulk sulfur is split into small particles, and PEG-400 is physically adsorbed on the surface of the sulfur powder, where the formed small particles can prevent the subsequent assembly process. With the extension of reaction time, the assembling effect competes with the fission effect, with the reaction being dominated by the assembling effect between 54 and 72 h and the fission effect between 72 and 125 h. After 125 h of reaction, the obtained SQDs are excitation-dependent, emitting green or blue light by adjusting reaction time, with QY enhanced to 3.8%.

### 2.3. Surface-Etching Method

Although Shen’s work (assembly-fission method) could improve QY, it is far from sufficient for fluorescent nanomaterials. In 2019, Wang reported an H_2_O_2_-assisted top-down approach to synthesize SQDs with a QY of 23.0% and the emission wavelength tunable from 440 to 500 nm [27]. The bulk sulfur powders are dissolved into small particles under an alkaline condition in the presence of PEG, followed by introducing H_2_O_2_ to etch the surface polysulfide species and transform the larger sulfur dots into smaller, size-controllable SQDs. Furthermore, researchers developed a type of negatively charged SQDs with QY of 5.1% by using poly (sodium 4-styrenesulfonate) (PSS) [28] as a capping agent instead of PEG-400 based on the H_2_O_2_-assisted surface-etching method (Figure 3). The surface-etching method has already become the most widespread and popular method for synthesis of SQDs.

Surface modification based on ligand exchange reaction has been shown to be effective in enhancing the QY of various nanocrystalline systems. Inspired by the H_2_O_2_-assisted surface-etching method, Li et al. [31] developed a facile method for the post-synthesis of SQDs, where Cu^2+^ was used as a precipitator to modify the prepared SQDs via a simple surface modification process. This post-synthesis Cu^2+^-assisted precipitation-etching method could obtain SQDs with a QY of 32.8%, but the biological application of these Cu-SQDs is restricted due to the potential toxicity of Cu^2+^.

### 2.4. Oxygen-Accelerated Method

In 2019, Song et al. [32] reported a simple, fast, and efficient approach for large-scale synthesis of highly fluorescent SQDs from inexpensive elemental sulfur powder through oxidation of divalent polysulfide (S_x_^2−^) ions to zero-valent sulfur under a pure oxygen (O_2_) condition. In their study, they found that O_2_ could accelerate the synthesis of SQDs by using sublimated sulfur, NaOH, and PEG-400 as experimental reagents under an O_2_ atmosphere (Figure 4a). The SQDs synthesized by this approach possess high fluorescence QY (21.5%), tunable emission, high stability against pH and ionic strength change, low toxicity, superior dispersibility, and long-term storage. Moreover, the content of S element in the SQDs reached 5.08%, much higher than that of previously reported methods (generally <1%).

However, the mechanisms of oxygen accelerated formation and photoluminescence of SQDs remain concealed to be fully elucidated. In 2021, Liu et al. [33] explored the mechanisms by O_2_ bubbling-assisted synthesis and spectroscopic analysis of SQDs produced from sulfur ions, which were formed from bulk sulfur with the passivation of PEG-400 under an alkaline condition (Figure 4b). The bubbled O_2_ is essential to the sulfur core and etching of its surface species. Experimental results demonstrated that the elemental sulfur core and the amount of surface divalent sulfur ions of SQDs dominate the emission color. Moreover, the emission color also depends on the size of SQDs. The luminescent intensity of SQDs is strongly affected by the surface divalent sulfur ions due to their quenching effect, which can be optimized by surface O_2_-oxidation to enhance the luminescent intensity. Liu’s work explained the function of O_2_ in the formation and luminescence intensity of SQDs, providing a guide for synthesis of strong luminescent SQDs.

Carboxymethyl cellulose (CMC) has been widely used to synthesize highly stable nanoparticles due to its advantages of excellent water solubility, abundant functional groups (such as -OH and -COOH groups), high biocompatibility, and low cost [43,44], suggesting it is more suitable than PEG for stabilizing SQDs. In 2020, Duan et al. [34] utilized CMC instead of PEG to synthesize SQDs using the oxygen-accelerated method. Specifically, fluorescent SQDs were obtained by stirring CMC, sublimed sulfur, and NaOH at 95 °C for 24 h under an O_2_ condition. The obtained fluorescent CMC-SQDs exhibited QY of 7.1%, high aqueous dispersibility and stability, tunable emission, and strong biocompatibility, conferring them the potential as a fluorescent probe. Similarly, hydroxypropyl-β-cyclodextrin (HP-β-CD) [35] and polyvinyl alcohol (PVA) [36] were also employed as ligands instead of PEG for synthesizing fluorescent SQDs with the oxygen-accelerated method. The obtained fluorescent SQDs showed high aqueous dispersibility, stable and tunable emission, and satisfactory biocompatibility.

### 2.5. Ultrasonication and Microwave Method

Despite the aforementioned progress in improving QY and the applications of SQDs, their sufficient applications are still limited due to lack of efficient and fast synthetic methods, requiring a long synthesis time of 125 h as reported by Shen et al. [30]. In 2019, Zhang et al. [37] proposed an ultrasonication-assisted method to promote the chemical etching of bulk sulfur into smaller particles through the reaction among sublimed sulfur powder, PEG-400, and sodium sulfide under sonication as shown in Figure 5. The reaction time was greatly shortened to less than 5 h, and the prepared fluorescent SQDs achieved QY of 2.1% with monodispersity, water solubility, and low cytotoxicity, paving the way for further reducing synthesis time to obtain highly fluorescent SQDs with wider applications. However, the QY of 2.1% for SQDs is far below the requirement of practical applications. In 2020, Hu’s group [38] developed the one-pot microwave-assisted strategy to synthesize luminescent SQDs with QY of 49.25% by using bulk sulfur powder and PEG-400 as reagents under an alkaline (NaOH) environment. Under microwave irradiation (5 min), rapid and uniform heat can be produced at a certain temperature; thus increasing the nucleation rate and adsorbing more sulfite groups, resulting in a relatively high QY after 40 h of treatment. The above analysis indicated that ultrasonication can efficiently shorten the synthesis time of SQDs, and their QY can be highly improved by microwave irradiation. In 2021, Sheng and co-workers [39] combined these two methods and proposed a one-pot synthesis of size-focusing fluorescent SQDs through ultrasound–microwave radiation, shortening the synthetic time to 2 h and achieving a record high QY of 58.6% and highly stable emission wavelength. Mechanistic studies indicate that the ultrasound–microwave heating procedure can provide an ultrafast heating rate, facilitating the rapid synthesis of SQDs by accelerating the etching process during the formation of size-focusing SQDs. The obtained SQDs possess a stable and single emission wavelength, with high QY and strong emission intensity.

### 2.6. One-Step Hydrothermal Method

The hydrothermal method has been widely used to synthesize various functional nanoparticles [45] due to its advantages of easy operation and low cost. In 2019, Xiao et al. [40] developed a strategy of hydrothermal reaction to synthesize SQDs, which could shorten the reaction time to 4 h by mixing bulk sulfur powder, PEG-400, and NaOH solution in a Teflon-lined autoclave chamber at 170 °C. The prepared fluorescent SQDs exhibited a QY of 4.02% without post-treatment. The fission-aggregation mechanism was proposed for the reaction dynamics during the formation of SQDs. Despite an obvious reduction in reaction time, this hydrothermal approach had a low QY. In 2021, Wang and co-workers [44] prepared for the first time the uniform and small-sized luminescent SQDs with QY of 10.3% through a one-pot solvothermal method. During synthesis, sublimated sulfur powder and PEG-400 were treated together in the presence of H_2_O_2_ at an elevated temperature (220 °C), where H_2_O_2_ acts as a dual-functional reagent, suspending the sulfur powder and etching bulk sulfur to construct uniform SQDs, while the elevated temperature (220 °C) promoted the formation of molten sulfur from sulfur powder as solvent, enhancing the reaction efficiency and production yield of SQDs. The as-prepared SQDs possess strong photoluminescence properties, excellent dispersibility, and high photostability.

### 2.7. Other Methods

Except for the synthesis of SQDs using the acid etching oxidation method (CdS QDs as precursor) in the first report by Li et al. [19], all the other methods were based on the top-down strategy, using bulk sulfur as the precursor. However, most of these methods require a long reaction time to synthesize fluorescent SQDs by etching bulk sulfur into small particles. In order to further the previous efforts, in 2020, Arshad et al. [42] reported the production of SQDs by using sodium thiosulfate (sulfur in S^2+^ state) as the precursor to form elemental sulfur in an in situ reaction with oxalic acid (Figure 6), followed by etching the as-formed elemental sulfur by NaOH under the passivation of PEG-400 to produce fluorescent SQDs. The obtained SQDs exhibited QY of 2.5%, excellent dispersibility, and high photostability.

The synthetic methods of SQDs are still under study and their further development is essential for the effective synthesis and sufficient applications of SQDs. Arshad’s group [26] undertook continuous research efforts on the synthesis of fluorescent SQDs within a shorter time. In 2021, for the first time, they developed the mechanochemical approach to synthesize SQDs through a short-chain polymerization of sulfur by using sodium thiosulfate as precursor, and the SQDs exhibited QY of 4.8%, low toxicity, high hydrophilicity, and superior photophysical properties.

## 3. Applications

### 3.1. Sensing

Despite their short research history, SQDs have been widely applied in sensing as mentioned in a few reviews [46,47,48]. Herein, we supplement the related achievements in the sensing application of SQDs in the past two years.

#### 3.1.1. Fluorescence Sensing

Since the first report of fluorescent SQDs in 2014 [19], with the continuous exploration of their synthetic methods, SQDs have been demonstrated to possess the advantages of excellent photoluminescence property, high QY, and excitation-dependent tunable emission. Moreover, similar to conventional semiconductor QDs, SQDs exhibited size-dependent photoluminescence emission [42,49]. These advantages resulted in the wide use of SQDs to develop fluorescent probes since the pioneering SQDs-based detection of Fe^3+^ [19].

Among the reported SQDs-based fluorescent sensors, some works are based on target regulation of the photoluminescence properties of SQDs, such as the aggregation-caused quenching of SQDs to detect Co^2+^ [50]. After adding norfloxacin, the fluorescence intensity of SQDs recovered, achieving a new detection strategy of norfloxacin. In 2020, Zhao et al. reported a turn on-off fluorometric assay for clioquinol (CQ) detection by using the Zn^2+^-CQ affinity pair to modulate fluorescence of SQDs [51]. In this work, Zn^2+^ acted as a fluorescence regulatory “bridge” to enhance the fluorescence of SQDs under a weak alkaline condition, and in the presence of CQ, the SQDs-Zn^2+^ fluorescence intensity was quenched, which increased the linear analytical range by two orders of magnitude and improved the selectivity of this method.

In recent years, a large number of fluorescent sensors have been designed based on the inner filter effect (IFE) mechanism [52]. In 2020, Li et al. [53] achieved ultrasensitive detection of butyrylcholinesterase (BChE) activity via the MnO_2_ nanosheet on SQDs based on IFE, where MnO_2_ nanosheet can effectively quench the fluorescence of SQDs, and BChE can catalyze its substrate to produce thiocholine; thus effectively converting MnO_2_ nanosheet into Mn^2+^, eliminating the IFE of MnO_2_ nanosheet on SQDs and restoring their fluorescence. The detection limit of this sensor is 0.035 U/L, with a two-stage linear relationship from 0.05 to 10 and from 10 to 500 U/L. In 2021, Tan et al. [54] reported that the fluorescence of SQDs could be quenched by Cr (VI) based on IFE due to the partial overlap of the excitation spectrum of SQDs and the absorption spectrum of Cr (VI). After introducing ascorbic acid (AA) into the SQDs-Cr (VI) system, the fluorescence intensity of SQDs could be recovered because of the reduction of Cr (VI) to Cr (III) by AA, achieving highly sensitive detection of Cr (VI) and AA, with detection limits of 0.36 and 1.21 μM, respectively. In the same year, Xia et al. [55] expanded the SQDs-based “on-off-on” fluorescent platform for detection of Cr (VI) and AA in real environmental and human samples, as well as Hela cells and zebrafish embryos/larvae (Figure 7). P-nitrophenol (p-NP) residues in soil and water environment are highly stable and toxic, resulting in serious damage to the ecosystem. In 2021, Peng et al. [56] realized the detection of p-NP in water samples based on the IFE between p-NP and SQDs, leading to the effective fluorescence quenching of SQDs. Additionally, Lu et al. [57] developed a novel fluorescent probe for sensitive and selective detection of tetracycline (TC) in milk based on SQDs, where fluorescence could be effectively quenched in the presence of TC due to the IFE mechanism.

The electron transfer (ET) effect is another mechanism commonly used to design fluorescence sensors. In 2021, Huang et al. [58] developed a selective fluorescence “on-off-on” sensor for detection of Fe^3+^ and phytic acid (PA) based on the introduction and elimination of the ET effect. The fluorescence of SQDs could be quenched in the presence of Fe^3+^ due to the non-radiative electron transfer between Fe^3+^ and SQDs, while the fluorescence of the SQDs/Fe^3+^ complex could be recovered after adding PA due to stronger binding affinity between PA and Fe^3+^. The detection limits for Fe^3+^ and PA are 102 and 73.5 nM, respectively. Meanwhile, Lei et al. [36] utilized SQDs as the fluorescence sensor for detection of intracellular Fe^3+^ based on the ET effect. In this work, PVA with abundant hydroxyl groups was carefully chosen as the ligand to prepare fluorescent SQDs for highly sensitive sensing of Fe3+ in water and in cells due to the strong complex interaction between Fe^3+^ and the hydroxyl groups with a detection limit of 92 × 10^−9^ M. Meanwhile, these PVA-capped SQDs showed highly reversible temperature-dependent fluorescence between 20 and 60 °C, and this behavior made them a nanothermometer to monitor cell temperature. However, the authors did not explain the mechanism for reversible temperature-dependent fluorescence of PVA-capped SQDs.

#### 3.1.2. Colorimetric and Fluorescence Dual-Channel Sensing

The fluorometric methods have been widely used for sensing various analytes due to their advantages of rapid response, excellent sensitivity, high selectivity, and simple operation [59]. However, fluorometric determination is based on single-signal systems, which are susceptible to external factors, such as background interference and environmental fluctuations [60]. Therefore, dual-signal determination systems are highly important in monitoring targets in real samples, such as the fluorometric/colorimetric dual-signal sensor system, which has been shown to self-calibrate the results for high accuracy. Moreover, the colorimetric method can be applied for in situ detection of analytes without any additional equipment.

In 2020, Qiao et al. [61] assessed the analytical performance of Hg^2+^ using a fluorometric and calorimetric dual-signal sensor based on SQDs. The fluorescence intensity of SQDs decreased with the addition of Hg^2+^ solution, and a linear relationship was achieved between F_0_/F and the concentration of Hg^2+^, with a fluorometric detection limit of 65 nM for Hg^2+^. Additionally, the authors found that the color of Hg^2+^ solution changed from transparent to a deep color, which could be obviously observed by the naked eye, and this observation was further confirmed by UV-Vis spectroscopic titration experiments, with a colorimetric detection limit of 1.86 μM for Hg^2+^. In 2020, Li et al. [62] developed a SQDs-based chemosensor for the fluorometric and colorimetric dual-signal detection of cobalt (Co^2+^) with good sensitivity and selectivity. Visual colors of this chemosensor in the presence of Co^2+^ varied obviously from blue to colorless under UV light irradiation and from colorless to yellow under sunlight. In this work, the fluorescence detection of Co^2+^ was based on the photo-induced electron transfer (PET) effect, with a detection limit of 0.16 μM for Co^2+^ (Figure 8). In 2022, Lu et al. [63] established a fluorometric and colorimetric dual-signal sensor for detection of iron (II) (Fe^2+^) and H_2_O_2_ based on SQDs. The fluorescence of SQDs could be quenched by Fe^2+^ due to the complexation between Fe^2+^ and SQDs, and the color of the mixture varied from light yellow to deep green. However, the quenched fluorescence could be recovered by H_2_O_2_ due to Fenton reaction between Fe^2+^ and H_2_O_2_, and the color of the mixture varied from green to colorless. The detection limit was 0.54 and 0.03 μM for Fe^2+^ and H_2_O_2_, respectively.

#### 3.1.3. Ratiometric Fluorescent Sensing

For improving the accuracy and anti-interference of biosensors, ratiometric sensing is another effective strategy by introducing another fluorescent material as a reference to record the fluorescence intensity ratio at two wavelengths [64,65]. Butyrylcholinesterase (BchE) is an important clinical diagnosis parameter, but its common detection protocol named Ellman’s colorimetry suffers from the problem of low sensitivity and interferences, pushing researchers to explore alternative strategies [66,67]. In 2021, Ma et al. [21] developed a ratiometric fluorescence sensor for detection of BchE activity based on the MnO_2_ nanosheet-modulated fluorescence of SQDs and o-phenylenediamine (OPD) (Figure 9). Specifically, the blue fluorescence of SQDs could be quenched and the yellow fluorescence of OPD could be promoted by MnO_2_ nanosheets simultaneously. However, MnO_2_ could be decomposed into Mn^2+^ after introducing BchE and its substrate to produce thiocholine, restoring the blue fluorescence of SQDs and inhibiting the production of yellow fluorescence of OPD. There was a linear relationship between the ratio of fluorescence intensity (F_435_/F_560_) and BChE in the concentration range of 30–500 U/L, with a detection limit of 17.8 U/L. In the same year, Zhuang et al. [25] developed a ratiometric fluorescent sensor for sensitive detection of doxycycline (Dox) in food based on SQDs and calcium ion (Ca^2+^). In this work, the fluorescence of SQDs at 450 nm could be effectively quenched by Dox due to static quenching and IFE. Meanwhile, a new fluorescence peak at 520 nm was produced due to the formation of Dox-Ca^2+^ complex through coordination. Therefore, the ratio of F_520_/F_450_ and Dox concentration showed a satisfactory linear relationship, with a detection limit of 0.19 μM. These ratiometric methods demonstrate the potential applications of SQDs in designing anti-interference sensors.

#### 3.1.4. Electrochemical Sensing

The electrochemical sensor has been widely explored due to its simplicity, sensitivity, convenience, and cheapness [68,69,70]. SQDs cannot conduct electricity, but they can affect the electrochemical signal of gold (Au) electrodes modified with SQDs, so they can be used to develop electrochemical sensors based on competitive response. In 2020, Fu et al. [71] proposed an electrochemical sensor for detection of silver ions (Ag^+^) based on an SQDs-modified Au electrode (Figure 10). In this work, SQDs could be modified on the surface of Au electrode due to the Au-S bond, leading to the decrease in the electrochemical signal of the Au electrode because of the poor electro-conductivity of SQDs. However, the SQDs/Au system exhibited a significant electrochemical response after introducing Ag^+^, and the strong affinity between Ag and S enabled the system to be highly sensitive for the detection of Ag^+^, with a detection limit of 71 pM.

#### 3.1.5. Electrochemiluminescence Sensing

Due to the combined characteristics of electrochemistry and chemiluminescence, ECL has developed into a powerful analytical technique for sensing and diagnostics [72,73]. ECL has shown the merits of facile controllability, low background, strong sensitivity, high selectivity, and wide response range [74]. In 2018, Shen et al. [30] found significant ECL signal from the prepared SQDs in an annihilation reaction, but they did not extend this finding to further application. Until 2020, Liu et al. [75] developed an SQDs-based (off-on) ECL biosensor with excellent ECL performance and an efficient DNA walking machine for miRNA-21 detection (Figure 11). In this work, the authors improved the ECL performance by optimizing the size and dispersity of SQDs, and the detection limit of this ECL sensor was 6.67 aM in the concentration range of 20 aM to 1 nM, opening the application of SQDs in the ECL field. In 2021, Han et al. [76] proposed a boosted anodic “off-on” ECL sensor based on high-quality SQDs (H-SQDs) for glutathione (GSH) detection. The ECL signal of the prepared SQDs was decreased by in situ assembly of MnO_2_ due to the ECL-resonance energy transfer system. However, the ECL signal could be recovered after introducing GSH to reduce MnO_2_ into Mn^2+^, leading to the release of H-SQDs. This ECL sensor showed excellent linearity in the range of 0.050–5.0 μM with the detection limits as low as 35 nM. This work promoted the development of ECL emitters, and more importantly, it provided a new avenue for exploring SQDs-based ECL biosensors in clinical diagnosis.

### 3.2. Bioimaging

Based on their good colloidal stability, tunable and stable optical properties, excellent aqueous dispersibility, low toxicity, and bioavailability, SQDs have been utilized as an attractive fluorescent agent for bioimaging applications. In 2019, Zhang et al. [37] first proved that SQDs can be applied as probes in live-cell imaging (Figure 12). In this work, BEAS-2B cells were incubated with SQDs in fresh medium, and intensive green emission could be observed in the cytoplasm of the BEAS-2B cells, indicating the cell imaging potential of SQDs. In 2020, Qiao et al. [61] investigated the bioimaging capability of SQDs. They demonstrated the potential of PEG-1000 passivated SQDs for imaging in two cell lines (HeLa and leukemia K562), and these SQDs were located in the cytoplasm after incubation with the cells for 8 h. Four sorts of chemical inhibitors were used to study the endocytosis mechanism of SQDs, and the results showed that the internalization processes of SQDs were mainly based on the clathrin and lipid-raft-mediated endocytosis mechanism. Moreover, cellular tracking of mercury ions was realized due to the specific fluorescence response between SQDs and mercury ions. Meanwhile, Song et al. [32] used fluorescent SQDs (20 μg/mL) to stain MCF-7 cancer cells for 2 h, and strong green and yellow emission could be observed in the cytoplasm under excitation at 458 and 514 nm, respectively, demonstrating the efficient internalization of SQDs by the MCF-7 cells. In 2021, Arshad et al. [26] synthesized fluorescent SQDs by mechanochemical method to incubate with Du-145 cells for 24 h at a concentration of 400 μg/mL, and they found that these as-prepared SQDs were highly internalized into the Du-145 cells and mainly located in both the cytoplasm and nucleus. In the same year, Wang et al. [41] investigated the cellular imaging of SQD-18 (3.0 mg/mL) incubating with HeLa cells for 24 h, and a bright blue fluorescence under 405 nm excitation was observed in the cytoplasmic area in the HeLa cells, suggesting that SQD-18 could penetrate cell membrane and label the cytoplasm. These studies all indicate a new insight that SQDs could be a promising application in cellular imaging fields.

## 4. Challenges and Prospects

In this minireview, we summarized the latest research progress of SQDs, including synthetic methods and sensing applications. Since the first report of using CdS as precursors in 2014, impressive progress has been achieved in the synthetic methods of SQDs, and due to their unique optical and physiochemical properties, SQDs have been applied in environmental and biological detection. However, compared to other QDs, the research of SQDs is still in the preliminary stage, and there are still many challenges and opportunities.

Currently, as shown in Table 1, for most reported synthetic methods, the synthesis time was shortened but with a relatively low QY, or QY was effectively improved but with a prolonged synthesis process. This suggested that more research efforts should be made to develop faster and simpler synthesis methods with higher QY and well-defined luminescence mechanisms. Meanwhile, the optical characteristics can be affected by side products, so the complex purification process should be simplified to achieve efficient product separation and purification.In these reported methods, most of the precursors were bulk sulfur powder, PEG, and NaOH; thus requiring a large amount of bulk sulfur powder. Although Arshad et al. [45] used sodium thiosulfate as precursors to produce elemental sulfur, that is not enough, and it is necessary to find new precursors to design effective reaction systems.From the perspective of sensing and bioimaging applications, one limitation for the reported SQDs is that most of their emission colors were focused on blue and green, which could be easily interfered with by the self-fluorescence of biological samples. This problem can be solved by synthesizing SQDs with red or near infrared or even infrared-II fluorescence, and such emission modulation can be achieved by doping heteroatoms, changing passivators and reaction conditions, etc.For sensing applications, the target molecules are limited, including metal ions (Fe^3+^, Ag^+^, Hg^2+^, Co^2+^, Zn^2+^, Cr^6+^, Ce^4+^), Dox, AA, CQ, norfloxacin, BChE, miRNA-21, and GSH. One of the possible reasons for this is the limited functional groups on the surface of SQDs, resulting in their limited ability to recognize the target molecules.

Overall, SQDs are promising nanomaterials, and once well studied, they can be applied in various fields. We hope that this minireview can provide researchers useful information to further the basic and applied development of SQDs.

## Data Availability

The data presented in this study are available on request from the corresponding author.

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
