# Peer review of "Research Update of Emergent Sulfur Quantum Dots in Synthesis and Sensing/Bioimaging Applications"

_molecules, 2022, doi:10.3390/molecules27092822_

Round 1
Reviewer 1 Report
The manuscript "Research update of emergent sulfur quantum dots in synthesis and sensing applications" by Ning K., Sun Y., Liu J., Fu Y., Ye K., Liang J. and Wu Y. is a very comprehensive review of the field of sulfur quantum dot research. Through good organization and the use of well-selected diagrams, it presents an update on the synthesis and sensing applications of this type of material, in a very appropriate way. I believe this review will be of value to the community specialized in materials research and their applications. I recommend this manuscript to be published in Molecules.
Author Response
The authors really appreciate the positive comment raised by the reviewer and thanks for the recognition and recommendation of our work.
Reviewer 2 Report
The minireview by Ning et al. provides a comprehensive update on recent advances in synthesis and sensing applications of non-toxic sulfur quantum dots. Overall, the review is well-structured and cites all the key publications in the field. Table 1 is excellent and provides a detailed list of available synthetic methodology towards sulfur QDs matched with the resulting optical properties and specific applications.
My only concern is that the first part of the review (Section 2. Synthetic Methods) is very similar to the review by Gao et al. on luminescent sulfur QDs (ChemPhotoChem 2020, 4, 5235–5244 = this paper is ref. 46 in the submitted manuscript). Specifically, Figures 1-4 in the submitted manuscript are the same as Figures 2-5 in the review by Gao et al., with the organization and flow of the material in this section generally the same. Could the authors reorganize this section and consider the use of different figures to boost novelty?
Author Response
Thank the reviewer for the valuable comments. Yes, some of the content in Section 2 in our manuscript is indeed a little similar to the content of ref. 46, this is because the synthetic methods discussed in these reviews are original innovative work and are discussed according to the chronological order of publication. We also discuss recently published synthetic methods, which are summarized in Table 1. Therefore, it is not suitable to reorganize this section. To highlight our differences from the ref.46, we have replaced the figure in Fig.3 as you suggested in our revised manuscript and added another figure as Fig.4b. In addition, Fig, 1 and Fig. 2 are not modified because both methods are only reported in the current literature.
Reviewer 3 Report
The manuscript entitled “Research update of emergent sulfur quantum dots in synthesis and sensing applications” summarized the recent approaches for the synthesis SQDs and used for sensing applications. The review article is well-written and the topic is of concern for the readers of Molecules; however, there are several suggestions as follows to improve its quality. Specific comments are listed below:
Comments:
- Page 2, Table 1: Add the full names of PEG, PSS, CMC, HP-b-CD, PVA, ECL, CL, LEDs, AA, and TTZ as footnotes in Table 1.
- Page 13, I suggest adding a section after section 3.5 to discuss the applications of SQDs for cell imaging (Ref. 26, 32, 39, 44). In addition, the SQDs in ref. 38 were used as a nanothermometer. It is also interesting. Introduce the content of ref. 38 in brief in the manuscript.
Author Response
Response: The authors really appreciate the kind and positive comment raised by the reviewer and thanks for the recognition of our work. We have carefully revised the manuscript to improve our paper quality according to your kind suggestion.
(Since we have revised the manuscript, the subtitle numbers in the manuscript also have been changed, the subtitle numbers mentioned in the “Responses to the reviewers” is the subtitle numbers in the revised manuscript.)
Question 1: Page 2, Table 1: Add the full names of PEG, PSS, CMC, HP-b-CD, PVA, ECL, CL, LEDs, AA, and TTZ as footnotes in Table 1.
Answer: Thank the reviewer for the helpful suggestion. We have added the full names of PEG, PSS, CMC, HP-b-CD, PVA, ECL, CL, LEDs, AA, and TTZ as footnotes in Table 1 in the revised manuscript in Page 3, Line 56-59. That is:
Abbreviations:
PEG: polyethylene glycol; PSS: poly (sodium-p-styrenesulfonate); CMC: carboxymethyl Cellulose, HP-β-CD: 2-Hydroxypropyl-β-cyclodextrin; PVA: polyvinyl alcohol; ECL: electrochemiluminescence; CL: chemilu-minescence; LEDs: light-emitting diodes; AA: ascorbic acid; TTZ: tartrazine.
Question 2: Page 13, I suggest adding a section after section 3.5 to discuss the applications of SQDs for cell imaging (Ref. 26, 32, 39, 44).
Answer: Thank the reviewer for the valuable suggestion. We have discussed the applications of SQDs in cell imaging in the revised manuscript. That is:
“3.2 Bioimaging
Based on their good colloidal stability, tunable and stable optical property, excellent aqueous dispersibility, low toxicity, and bioavailability, SQDs have been utilized as an attractive fluorescent agent for bioimaging applications. In 2019, Zhang et al. [39] first proved that SQDs can be applied as probes in live-cell imaging. In this work, BEAS-2B cells were incubated with SQDs in fresh medium, and intensive green emission could be observed in the cytoplasm of the BEAS-2B cells, indicating the cell imaging potential of SQDs. In 2020, Qiao et al. [61] investigated bioimaging capability of SQDs. They demonstrated the potential of PEG-1000 passivated SQDs for imaging in two cell lines (HeLa and leukemia K562), and these SQDs were located in the cytoplasm after incubation with the cells for 8 h. Four sorts of chemical inhibitors were used to study the endocytosis mechanism of SQDs, and the results shown that the internalization processes of SQDs were mainly based on the clathrin and lipid raft mediated endocytosis mechanism. Moreover, cellular tracking of mercury ions was realized due to the specific fluorescence response between SQDs and mercury ions. Meanwhile, Song et al. [32] used fluorescent SQDs (20 μg/mL) to stain MCF-7 cancer cells for 2 h, and strong green and yellow emission could be observed in the cytoplasm under excitation at 458 and 514 nm, respectively, demonstrating the efficient internalization of SQDs by the MCF-7 cells. In 2021, Arshad et al. [26] synthesized fluorescent SQDs by mechanochemical method to incubate with Du-145 cells for 24 h at a concentration of 400 μg/mL, and they found that these as-prepared SQDs were highly internalized into the Du-145 cells and mainly located in both the cytoplasm and nucleus. In the same year, Wang et al. [44] investigated the cellular imaging of SQD-18 (3.0 mg/mL) incubating with HeLa cells for 24 h, and a bright blue fluorescence under 405 nm excitation was observed in the cytoplasmic area in the HeLa cells, suggesting that SQD-18 could penetrate cell membrane and label the cytoplasm. These all studies indicate a new sight that SQDs could be as a promising application in cellular imaging fields.”
Question 3: In addition, the SQDs in ref. 38 were used as a nanothermometer. It is also interesting. Introduce the content of ref. 38 in brief in the manuscript.
Answer: Thank the reviewer for the valuable suggestion. We have supplemented a brief introduction of ref. 38 in the revised manuscript in Page 10, Line 287-294. That is:
“In this work, PVA with abundant hydroxyl groups was carefully chosen as ligand to prepare fluorescent SQDs for highly sensitive sensing of Fe3+ in water and in cells due to the strong complex interaction between Fe3+ and the hydroxyl groups with a detection limit of 92×10−9 M. Meanwhile, these PVA-capped SQDs showed highly reversible temperature-dependent fluorescence between 20 and 60 °C, and this behavior made them as a nanothermometer to monitor cell temperature. However, the authors did not explain the mechanism for reversible temperature-dependent fluorescence of PVA-capped SQDs.”
